# Salt Tolerance in Soybeans: Focus on Screening Methods and Genetics

**DOI:** 10.3390/plants13010097

**Published:** 2023-12-28

**Authors:** Rong-Xia Guan, Xiao-Yang Guo, Yue Qu, Zheng-Wei Zhang, Li-Gao Bao, Rui-Yun Ye, Ru-Zhen Chang, Li-Juan Qiu

**Affiliations:** 1The National Key Facility for Crop Gene Resources and Genetic Improvement (NFCRI), Key Lab of Soybean Biology, Ministry of Agriculture, State Key Laboratory of Crop Gene Resources and Breeding, Institute of Crop Sciences, Chinese Academy of Agricultural Sciences, Beijing 100081, China; guoxiaoyanggxy@163.com (X.-Y.G.); 13121233799@163.com (Z.-W.Z.); rx_guan@sina.com (R.-Z.C.); 2Australian Research Council Centre of Excellence in Plant Energy Biology, School of Agriculture, Food and Wine, Waite Research Institute, University of Adelaide, Glen Osmond, SA 5064, Australia; yue.qu@adelaide.edu.au; 3Agriculture and Animal Husbandry Technology Promotion Center of Inner Mongolia Autonomous Region, Hohhot 010018, China; nmgnmyjs@163.com; 4The Economic Development Center of China State Farm, Beijing 100122, China; guochan2669@163.com

**Keywords:** soybean, salinity stress, salinity tolerance, ion homeostasis, gene identification

## Abstract

Salinity greatly affects the production of soybeans in arid and semi-arid lands around the world. The responses of soybeans to salt stress at germination, emergence, and other seedling stages have been evaluated in multitudes of studies over the past decades. Considerable salt-tolerant accessions have been identified. The association between salt tolerance responses during early and later growth stages may not be as significant as expected. Genetic analysis has confirmed that salt tolerance is distinctly tied to specific soybean developmental stages. Our understanding of salt tolerance mechanisms in soybeans is increasing due to the identification of key salt tolerance genes. In this review, we focus on the methods of soybean salt tolerance screening, progress in forward genetics, potential mechanisms involved in salt tolerance, and the importance of translating laboratory findings into field experiments via marker-assisted pyramiding or genetic engineering approaches, and ultimately developing salt-tolerant soybean varieties that produce high and stable yields. Progress has been made in the past decades, and new technologies will help mine novel salt tolerance genes and translate the mechanism of salt tolerance into new varieties via effective routes.

## 1. Introduction

Soil salinity is due to the accumulation of soluble salts, including chlorides and sulfates of sodium, and more than 3% of farmlands are seriously threatened by salinization [1,2]. Salinity usually inhibits crop growth through the osmotic effect, which reduces a plant’s water take up or ion toxic effect, which inhibits enzyme activity [3]. The extent and severity of the effect of saline soils on crop production is predicted to worsen because of factors such as global warming and inadequate drainage of irrigated land [4,5,6]. The need to expand agriculture into marginal lands, coupled with increasing global food requirements due to increasing population sizes, requires the development of crops that can achieve higher yields in soils with higher salt contents [7,8]. Salt stress reduces the yield of most crops [4,9,10,11]. The exploration of salt-tolerant crop plants that can withstand high salinity is considered one of the most effective biological strategies to cope with this problem and sustain food production [12]. Among the wide range of salt concentrations in saline soils, it was found that moderately saline soil had a minimal effect on reducing the soybean seed yield of salt-tolerant varieties, indicating the promising potential for using these varieties in saline fields [13].

As a source of vegetable protein and oil worldwide, soybean (*Glycine max* L. Merrill) plays important roles in human nutrition, animal feed, and oilseed production [14]. It was reported that extracts of soil solutions with conductivity (ECe) values of 5 dS m^−1^ or greater affect soybean germination and later developmental stages [15,16]. Soil salinity values of 7.3 and 9.6 dS m^−1^ caused complete stand loss of salt-sensitive and intermediate salt-tolerant soybeans, respectively. In contrast, the stands of salt-tolerant varieties were not appreciably decreased with salt, indicating inheritance control over salt tolerance [13]. Significant progress has been made in the evaluation and genetic analysis of soybean salt tolerance during the past few decades. In this review, we focus on the advances in genetics and possible mechanisms of salinity tolerance in soybeans at different growth stages. Suggestions are made for future studies that aim to improve salt tolerance via the ontogeny of soybeans.

## 2. Identification of Salt-Tolerant Accessions in Soybean

### 2.1. Salt Tolerance Evaluation at Soybean Germination Stage

Many germination tests have illustrated the negative effects of salt exposure on soybean germination. High salt concentrations decrease the soybean germination rate and inhibit the growth of radicle and lateral roots [17]. A study conducted in 1983 provided genetic resources on soybean salt tolerance at the germination stage and identified salt-tolerant soybean landraces and varieties under controlled conditions. The researchers used the relative injury index as a criterium to score the salt tolerance of soybean accessions from level 1 (a relative injury index of 0–20%) to level 5 (90.1–100%) at high salinity (a 1.6% NaCl solution) and were able to select tolerant soybean genotypes with scores of lower than 3 (a relative injury index of lower than 65%) [18]. In the Seventh Five-Year Plan of China, a total of 10,128 soybean accessions were evaluated for salt tolerance at the germination stage, and the results showed that 9.1% of genotypes were tolerant (based on relative injury index levels 1–3) [19]. Parameters, such as germination percentage (G%), tissue water content (TWC), and root length (TL), were considered useful indicators for the selection of salt-tolerant soybean in the germination stage while exposed to 150 mM and 200 mM NaCl stress [20]. The relative germination index (ST-GI) and relative germination rate (ST-GR) were significantly positively correlated with each other under 150 mM NaCl stress and could be used as indexes of salt tolerance at the germination stage [21].

### 2.2. Evaluation of Salt-Tolerant Soybean Accessions at Emergence Stage

Salt tolerance at the emergence stage is likely to be more important than in the germination stage because germinated seeds under certain saline conditions, like germination on salinized filter paper, may not break through the soil crust when the soil surface is hard [22]. Salinity decreased the emergence rates of soybeans grown in soil with less than 1.0% salt (a dry soil base), despite all soybeans achieving maximum germination rates. Differences in relative emergence rates were observed between the different soybean varieties grown under higher salt contents, which might be due to the effect of chloride ions [13,23]. At a salt concentration of 330 mM, the soybean cv. Williams attained a high germination rate (81%); however, seedling growth declined to 5%, even when exposed to a lower 220 mM NaCl stress [23]. This suggests that soybean seeds can survive under saline conditions during the germination stage, but the stress can be fatal at the emergence stage. Only one soybean genotype was used in this experiment; therefore, a variety of soybean accessions should be used to evaluate the correlations between genotypic differences at the germination and emergence stages. Variations in salt tolerance were reported in near-isogenic lines (NILs) that differ in maturity in the background of the soybean cultivars ‘Lee’ and ‘Essex’, indicating that specific genes in the NILs of maturity groups IV or VI may related to salt tolerance [24]. Recently, two flowering-related loci, *E2* and *J*, were found to be related to soybean seedling salt tolerance. Knockout of *E2* generated soybean lines with enhanced salt tolerance and shortened maturity [25], and loss of the function of *J* reduced salt tolerance and prolonged the maturity of soybean [26], indicating that flowering earlier might help soybeans avoid salinity toxicity. However, the effects of maturity-related genes on germination under salinity stress remain unknown. In fields salinized with sodium chloride and calcium chloride salts, soybean emergence was significantly reduced when the ECe reached 11 dS m^−1^ [24]. In order to overcome the spatial heterogeneity in saline field trials, Liu et al. (2020) developed a method using vermiculite as a culture substrate treated with a salt solution after soybean sowing, whereby soybeans grown under 150 mM NaCl showed a different salt tolerance. The salt tolerance index (SI) calculated using the relative growth of seedlings was found to be significantly related to salt tolerance and was used for the identification of salt-tolerant soybean accessions [27].

### 2.3. Salt Tolerance of Soybean at Seedling Stage

The screening of salt tolerance at the different stages of soybean growth has been conducted for more than half a century (Table 1). Much of the research has focused on soybean seedling stages because salt tolerance increases with the progression of plant age [18]. Much effort has been devoted to developing rapid, visual methods for the selection of salt-tolerant accessions. One such effort categorized soybeans as “includers” (salt-sensitive) or “excluders” (salt-tolerant). Soybeans grown under saline conditions accumulate more chloride ions in the stems and leaves, which causes severe leaf necrosis and even mortality, and were thus assigned as “includer” soybeans. Those soybeans that showed no necrosis and only a moderate growth reduction were assigned as “excluder” soybeans [28]. Three Cl excluders and four includers were grown hydroponically with 0, 40, 80, 120, and 160 mM NaCl for seven days. Includers and excluders showed the greatest differences at the 120 mM NaCl stress level, where the average leaf Na^+^ and Cl^−^ contents were 2.64 and 1.96 times higher for includers than excluders, respectively. Thus, the addition of 120 mM NaCl in the hydroponic system was the most effective concentration for screening salt-tolerant soybean genotypes without chemical analysis [29]. A method using sandy soil in plastic containers (named the PC method) was compared with the hydroponic method by exposing 14 soybean genotypes to salt at the V2–V3 stages. The leaf scorch scores and leaf chloride contents were comparable for both methods and salt damage appeared approximately four days sooner in the PC method; thus, the cheaper and less labor-intensive PC method was considered the better one that could be adopted by soybean breeders [30]. Using two Cl includers and two excluders as materials, a pot assay of soybean exposed to 120 mM NaCl over two weeks (for 2 h in a salt solution each day) was recently conducted to establish a leaf scorch scale (LSS) to visually rate the level of salt tolerance of plants. The scale ranges from one (healthy dark green leaves with no chlorosis) to nine (necrotic leaves). The results indicate that the LSS was positively correlated with Cl^−^ contents in leaves (*r* = 0.87–0.88; *p* < 0.001), which suggests that the LSS and Cl^−^ content may be used as criteria to identify tolerant genotypes [31]. In a simple screening method using vermiculite as a substrate, 200 mM NaCl was added to the tray every two days over five days (totaling 600 mM NaCl) after soybean unifoliate leaves were fully expanded, and a significant negative relationship of the leaf chlorophyll content (SPAD value) with the leaf Na^+^ content and salt tolerance was observed eight days after the final NaCl solution application [32]. This simple method was effectively used in salt tolerance gene mapping and functional analysis [33,34,35]. In a pot assay, a 25 mM NaCl solution was added to the soil every alternate day until a total concentration of 150 mM NaCl was added to each pot. A total of 170 soybean accessions were assigned to four groups: tolerant, moderately tolerant, moderately sensitive, and sensitive accessions. The relative total dry weight (DW) of 30-day-old seedlings, followed by the relative shoot and petiole DW, were considered as the more important discriminatory variables, while the relative root DW was considered as a secondary variable used to segregate accessions into the four groups [36]. This is in agreement with a previous study that reported the shoot growth of soybeans was more affected than the root growth under saline conditions [37]. In contrast, Lee et al. (2008) [30] observed that the shoot dry weight was less affected than the root dry weight. The conflicting results may be attributed to the experimental methods and mediums as well as genotypes used in each study. Overall, the comparisons suggest that relative dry weight is an unreliable indicator to assess salt tolerance in plants, and screening methods based on leaf injury have been widely used for salt tolerance evaluation at the seedling stage. To better understand the genetic control of salt tolerance, efforts should be spent on developing effective screening methods that mimic saline conditions in the field. Moreover, precautions must be taken to avoid common issues in large-scale evaluations of the salt tolerance of soybeans in the field when irrigating large fields with saline water [13,19]. For example, soil salinity can greatly vary from the head- to the tail-ends of furrow-irrigated fields and negatively affect the precision of evaluating plant salt tolerance [24,38].

### 2.4. Salt Tolerance Identification in Later Soybean Growing Period

Salt stress affects many agronomic traits of soybean, especially yield, mainly via the reduction in the branch number, pod number, grain weight, and 100-grain weight, which ultimately leads to yield reduction [42]. Screening soybeans throughout the whole growing period in a saline field is difficult, especially for a large number of accessions, due to salinity heterogeneity and uncontrolled environments. Saline water irrigation in field or pot experiments was used for the identification of salt tolerance at later soybean growing stages. Soybean accessions were planted in saline fields and irrigated with saline water (EC = 20 to 24 dS m^−1^) at the flowering and podding stages, and seven tolerant varieties like Wenfeng 7, Jindou 33, and Tiefeng 8 were identified [18]. Using salt-tolerant varieties (Wenfeng 7, Zhongye 1, Tiefeng 8, and Zhonghuang 10) as controls, a two-year salt tolerance evaluation of 280 soybean varieties was conducted in saline fields during the whole growth period of soybean, the biomass and grain weight of each variety were investigated at the maturity stage, the relative salt tolerance index was calculated, and only 35 soybean varieties showed high tolerance in both years, in which only 3 and 11 were tolerant at the germination and seedling stages, respectively [41]. In a potting experiment, soybean varieties were treated with 80 mM NaCl at the V3, R2, R4, and R6 stages, respectively. Biomass and pod weight were greatly affected in both tolerant and sensitive genotypes when salt was applied at the R6 stage, indicating that R6 is one of the most sensitive stages to salinity stress [43].

## 3. Genetics of Salinity Tolerance in Soybeans

### 3.1. Genetic Control of Salt Tolerance in Soybeans

Variations in salt tolerance exist in soybeans at different developmental stages, indicating potential tolerance gene resources in soybean accessions. To understand the genetic architecture of salt tolerance and improve selection efficiency, linkage and association analyses were used to identify loci controlling salt tolerance at the germination stage. Indices (the relative imbibition ratio (IR); the relative ratio of germination index (GI]); and the relative germination rate (GR)) representing possible mechanisms were used, and 11 QTLs located on chromosomes 2, 7, 8, 10, 17, and 18 underlying the complex traits were identified in an RIL population, NJRIKY, developed from a cross between Kefeng 1 and Nannong 1138-2 [21]. In the association analysis of natural soybean populations, 11 SNP- and 22 SSR-trait associations were identified [21,40]. Although several candidate genes on chromosomes 8, 9, and 18 were verified in response to salt stress, no consistent loci were identified in bi-parental segregating and natural populations [21]. The significant correlation between different indices, such as the GI and GR, and the co-association with related markers indicated that effective indices like the ratio of the germination rate under salt conditions to the germination rate under no-salt conditions (ST–GR) can be used for future experiments. The advantage of using the GR over the GI is that to obtain the GI, we need to manually count the number of germinated seeds in each Petri dish daily throughout the experiment, while to obtain the GR, we only need to count the number of germinated seeds once, at the end of the experiment. In an association mapping study of salt-tolerance-related markers in the emergence stage, the salt tolerance index (STI) based on the phenotypes of root length (LR), fresh or dry root weight (FWR; DWR), the biomass of seedlings (BS), and the length of hypocotyls (LH) were used as indices, and 19 QTLs were detected on various chromosomes, with only 2 related to LR-STI. Additional results showed that epistatic interactions between QTLs related to FWR-STI had strong effects (*r*^2^ > 5%) [44]. The minor effects and fewer conserved QTLs in different populations indicated that salt tolerance at the germination and emergence stages are likely controlled by quantitative loci.

In order to investigate the potential genetic control of salt tolerance, crosses were made using soybean chloride includers and excluders and 30-day-old seedlings of segregating populations grown in fields that were furrow-irrigated with saline water (an equal mixture of NaCl and CaCl_2_). The F_2_ populations of the includers × excluders were segregated in ratios of three non-necrotic plants (low chloride content) to one necrotic plant (very high chloride content), indicating the likelihood of a single gene governing the inheritance of salt tolerance. The gene symbols *Ncl* and *ncl* were proposed for the dominant (excluder) and recessive (includer) alleles, respectively. The phenotype of individuals was primarily sorted according to the level of leaf necrosis, while only 10 individuals were determined by the chloride concentration [28]. More than two decades later, three salt-tolerant and three salt-sensitive soybean cultivars were used to make crosses and evaluated for salt tolerance inheritance. Populations were planted in saline fields and irrigated with saline water (EC = 4.2 to 21 dS m^−1^, depending on current drought or non-drought conditions). The resulting segregation of phenotypes indicated that salt tolerance in all of the three tolerant cultivars was governed by a dominant gene [45]. In the F_2:3_ population derived from the cross of Peking (salt-sensitive) and wild soybean NY36-87 (salt-tolerant), the ratio of salt-tolerant to separation to salt-sensitive families was consistent with 1:2:1, indicating that the seedling salt tolerance of the NY36-87 wild soybean is controlled by a dominant single gene [46].

An F_2:5_ population from a cross of the salt-tolerant cultivar S-100 and the salt-sensitive cultivar Tokyo, for the first time, was used for salt tolerance QTL mapping. The heritability of salt tolerance was 0.85 and 0.48 in the field and greenhouse environments, respectively. A major salt tolerance QTL was mapped on the soybean linkage group N (LG N, Chr. 03) in a 3.6 cM interval between SSR markers Sat_091 and Satt237 based on the phenotypes of plants grown in a field, greenhouse, and combined environments of the two [47]. This major QTL was confirmed in several salt-tolerant soybean varieties, such as Nannong 1138-2, Tiefeng 8, Jidou 12, Fiskeby III, and FT-Abyora [33,48,49,50,51]. The genetic and mapping results have been described in more detail in recently published reviews [15,52,53,54,55]. The leaf sodium (LSC) and leaf chloride (LCC) contents of the F_2:3_ population derived from Williams 82 and the tolerant soybean Fiskeby III were used as physiological traits in QTL mapping. For the LCC, only one genomic region with a high *R*^2^ (58.9%) was identified on Chr. 03, where the major salt tolerance QTL was located. While for the LSC, except for the locus on Chr. 03, another dominant gene (a positive allele in the sensitive parent Williams 82) was located on Chr. 13, which explained 11.5% of the observed total variation, and no significant epistatic interactions were detected between these two loci [51]. Genome-wide association mapping based on leaf chloride concentration and SPAD showed SNPs on Chr. 02, 03, 14, 16, and 20, and all were significantly associated with both traits. These results suggest novel genes are involved in soybean salt tolerance at the seedling stage and the potential application of these SNP markers in the evaluation of accessions and breeding selection [56]. Recently, two major QTLs associated with the LSS and CCR were identified in Williams82 × PI483460B RIL populations. *qSalt_Gm03*, associated with the CCR and LSS, was located in the same region as the known salt tolerance gene *GmCHX1*. Another new locus, *qSalt_Gm18*, significantly associated with the LSS, was mapped on Chr. 18. The salt tolerance alleles of the two loci were both from PI483460B [57].

Salt tolerance gene mapping of wild soybeans has also obtained vital progress in addition to that of cultivated soybeans. In F_2_ populations derived from soybean cultivars and salt-tolerant wild soybeans, a major salt tolerance QTL was mapped on Chr. 03 [46,58], indicating that the same QTL or major gene was present on Chr. 03 in wild and cultivated soybeans (Figure 1). An F_2_ population of PI483463 × S-100 was used to determine the allelic relationship of wild accession PI483463 and cultivar S-100. The population was segregated as 15 (tolerant):1 (sensitive), indicating that the gene in wild soybean was different from that in S-100, and the gene was assigned as *Ncl2* in PI483463 [59]. However, the salt tolerance QTL in PI483463 was mapped within a 658 kb region on Chr. 03 using an RIL population derived from PI483463 and Hutcheson [60]. Because different sensitive parents were used in these two studies and PI483463 showed a higher tolerance than S-100 after 30 days of salt stress, it is difficult to rule out the possibility that PI483463 has a different salt tolerance gene [59,60]. A new salt tolerance locus on Chr. 18, named *GmSALT18*, was identified in an F_2:3_ population derived from the salt-sensitive variety Peking and the salt-tolerant wild soybean NY36-87 [46]. These wild soybeans should be further investigated to clone novel salt tolerance genes.

### 3.2. Candidate Gene Contributes to Salt Tolerance in Soybean

A major salt tolerance locus *qST-8* related to salt tolerance at the soybean germination stage was mapped onto Chr. 08 using QTL mapping in the RIL population and GWAS in the natural population. *Glyma.08g102000*, which belongs to the CDF (cation diffusion facilitator) family, was found to be the candidate gene of *GmCDF1*. Hairy root transformation experiments showed that the gene negatively regulated soybean salt tolerance by maintaining K^+^–Na^+^ homeostasis in shoots under salt stress. Haplotype analysis showed that two SNPs were significantly associated with salt tolerance, and Hap2 was more salt-tolerant than Hap1 (Table 2) [61].

The major salt tolerance QTL locus on Chr. 03 was conserved in both cultivated and wild soybeans. The isolation of the dominant gene has been the focus of extensive research efforts. By re-sequencing 96 RI lines derived from the salt-tolerant wild soybean W05 and the sensitive cultivar C08, a bin map was constructed, and a salt-related QTL was mapped in a 388 kb genomic region that overlapped with a previously mapped *Ncl* locus on Chr. 03. A root-specific expressed cation/H^+^ exchanger gene *Glyma03g32900* was identified as the candidate gene, which was named as *GmCHX1*. The expression of *GmCHX1* in hairy roots leads to a higher fresh root weight than the control. Moreover, transgenic tobacco BY-2 cells showed higher survival rates under 100 mM NaCl treatment, confirming the salt tolerance function of this candidate gene was from wild soybean (Table 2) [62]. A map-based cloning strategy was conducted for fine mapping the salt tolerance gene *GmSALT3* (a salt-tolerance-associated gene on chromosome 3) in cultivated soybean Tiefeng 8, and only *Glyma03g32900*, an endoplasmic-reticulum-localized gene, was predicted to be present in a 17.5 kb candidate region according to the reference genome Williams 82 (Table 2) [34]. Salt-tolerant wild soybeans were used for the identification of a novel salt tolerance gene, and a 7 bp InDel in the promoter region of *Glyma.11G149900* (*GsERD15B*) was found to be associated with salt tolerance. Genetic transformation proved that a Hap2-type promoter enhanced hairy root growth under salt stress, and 87.5% (42 of 48) of tolerant soybeans belong to Hap2. The average STR (salt tolerance rating) of Hap1 is 4.20, which is significantly higher than that of Hap2 (1.64) (Table 2) [63].

In addition to forward genetics, a series of genes encoding ion transporters and transcriptional factors were cloned from soybean via homologous cloning and functionally evaluated in *Arabidopsis*, tobacco, or soybean [64,65,66,67,68,69,70]. Functionally verified salt tolerance genes in soybeans have been carefully summarized in a recent review [71].

**Table 2 plants-13-00097-t002:** Salt tolerance genes identified in soybean and related molecular markers.

Tolerance Gene	Associated Markers	Salt Tolerance	Reference
*GmSALT3*	Pro-Ins, H2-Ins, H3-MboII, H4-NlaIII, and H5-Del	Seedling stage	Guan et al. (2021) [72]
	Tn-I, I-S, TGCT-D, and C-I	Seedling stage	Lee et al. (2018) [73]
*GmCHX1*	—	Seedling stage	Qi et al. (2014) [62]
	M1, M2, M3, M4, and M5	Seedling stage	Patil et al. (2016) [74]
*GmCDF1*	—	Germination stage Seedling stage	Zhang et al. (2019) [61]
*GsERD15B*	dCAPS-*GsERD15B*-promoter	Seedling stage	Jin et al. (2021) [63]

### 3.3. Dissecting Salt Tolerance Mechanisms in Soybean

Mechanisms related to the salt stress response, including signaling, osmotic stress, and ionic homeostasis, have been reviewed in detail [71]. In this review, we focus on salt tolerance genes cloned using forward genetics, because these genes are likely to be more valuable in marker-assisted selection breeding. However, this does not preclude that other genes may also contribute to salt tolerance breeding.

Independent studies have cloned the major salt tolerance gene *GmSALT3*/*GmCHX1*/*GmNcl*, which regulates salt tolerance at the soybean seedling stage [34,62,75]. GmSALT3 is an ER-localized protein regulating ions transport to shoots in a root-dependent manner [34]. The salt tolerance gene *Ncl* can reduce Na^+^, K^+^, and Cl^−^ accumulation in soybean leaves under salt stress and function like cation–chloride cotransporter (CCC) genes [75]. Recently, it was proved in a heterologous system that the GmSALT3 protein contributed Na^+^, K^+^, and Cl^−^ transport in *Xenopus laevis* oocytes. Detailed analysis of three sets of salt-tolerant NILs (NIL-*GmSALT3*) and salt-sensitive NILs (NIL-*Gmsalt3*) showed that *GmSALT3* mediates Na^+^ and Cl^−^ exclusion from shoots via net xylem loading or phloem re-translocation, although the exact molecular mechanism requires further study [35,76]. Transcriptomic analysis has been used to unravel the molecular mechanisms of *GmSALT3*, which suggests GmSALT3 might help to detoxify ROS toxicity through the flavonoid biosynthesis pathway [77]. Recently, it was reported that the membrane-bound NAC with trans-membrane motif1-like (NTL) transcription factor GmNTL1 can bind to the promoter of *GmSALT3*/*GmCHX1*/*GmNcl* to promote soybean salt tolerance by activating gene transcription [78], while the other gene, *GmERD15B*, might promote soybean salt tolerance via the up-regulation of stress-related genes, including *GmbZIP1*, *GmP5CS*, *GmCAT4*, and *GmSOS1* [63].

Salinity inhibits seed germination in plants by altering different growth processes including the imbibition of water, enzyme activities, and hormonal balance [3]. *GmCDF1* is the only candidate gene related to salt tolerance at the germination stage, the function of which was proved in soybean hairy roots. Overexpression of *GmCDF1* could decrease soybean salt tolerance by affecting K^+^–Na^+^ homeostasis in soybean roots and shoots [61]. The mechanism through which *GmCDF1* regulates soybean germination under salt stress needs further exploration.

## 4. Salt-Tolerant Accessions and Molecular Markers in Soybean Breeding for Saline Soils

### 4.1. Major Genes Involved in Soybean Salt Tolerance Provide Molecular Approach to Tolerant Soybean Screening and Breeding

An RAPD marker tightly linked to the salt tolerance gene was identified in the soybean cultivar Jindou 33 and used for the identification of tolerant accessions [79,80]. Subsequent sequencing of specific fragments of the RAPD marker in soybeans showed that the sequence is part of *Glyma03g32920.1*, from which a sequence-characterized amplified region (SCAR) marker was developed for the fine mapping of the salt tolerance gene [33]. Two SSR markers, Sat_091 and Satt237, were suggested to be useful for salt-tolerant soybean breeding due to the tight linkage of SSR markers with major QTLs on linkage group N and the association of marker alleles with salt tolerance in soybean descendants [47]. With the cloning of the major salt tolerance gene *GmSALT3*/*GmCHX1*/*GmNcl* [34,62,75], variations in the gene promoter and coding regions were identified in diverse soybean accessions [72,74]. Taking those haplotypes observed in more than ten soybean accessions as the main haplotypes, there are seven main haplotypes of *GmSALT3*/*GmCHX1*/*GmNcl* reported in soybean (Figure 2). Haplotype H1/Hn/SV-1 is the only conserved tolerant allele that has a functional domain. H2/HTn/SV-2 is a salt-sensitive haplotype with a 3.78 kb Ty1/copia retrotransposon insertion in the third exon; H5-1/Hd-2 is a sensitive haplotype with a 4 bp deletion in exon 2; and H5-2/Hd-3 has the same 4 bp deletion as that of H5-1/Hd-2, with an additional C > G variation in exon 3. H3 and H4 are two sensitive haplotypes only observed in Chinese soybean accessions [34,72]. SV-3 is a sensitive haplotype with a ~180 bp deletion in exon 3, and this allele was only reported by Patil et al. [74]. SNP assays and PCR-based markers were developed according to the variations in *GmSALT3*/*GmCHX1* and showed the precise identification (>90%) of salt-tolerant accessions, providing functional markers for targeted breeding [72,73,74]. A new variation in the promoter region of *GmCHX1* was proved to be a conditional gene-expression-related allele that existed in four salt-tolerant lines. It provides a new allele for salt tolerance breeding [81]. A 7 bp Indel in the promoter region of an early responsive to dehydration 15B (*GsERD15B*) gene was found to be related to the salt tolerance rating in wild soybean (*G. soja*), and a dCAPS marker was designed to distinguish the two alleles [63]. It remains unknown what the variation of *GsERD15B* in cultivated soybeans is and how it can be used in soybean breeding. Ten haplotypes of *GmCDF1* were detected that control salt tolerance at the germination stage, and haplotype Hap2 was more tolerant than Hap1 [61], while no further molecular markers related to this gene were reported.

### 4.2. Creation of Salt-Tolerant Soybean

The identification of salt tolerance loci is likely to contribute to the development of salt-tolerant soybean varieties. To confirm the function of known tolerance genes like *GmSALT3*, near-isogenic lines harboring *GmSALT3* (NIL-T) or *Gmsalt3* (NIL-S) were created using a marker-assisted strategy. Each pair of NILs contained 95.6–99.3% genetic similarity and were used to elucidate gene function in salinized soil. No yield penalty was observed for *GmSALT3* under normal field conditions, and a significantly higher 100-seed weight and total plant seed weight were found in NIL-T lines in salinized fields [35]. Under salt stress, NILs with the salt tolerance allele showed a yield decrease of less than 29.5%, whereas NILs with the salt sensitivity allele experienced a more pronounced yield reduction of 44.0–55.8%, indicating the presence of the salt tolerance gene contributed to sustainable soybean production in saline fields [75]. Commercial soybean cultivars containing *GmSALT3*, such as Zhonghuang 30 and Zhonghuang 13, which have been approved to be salt-tolerant, are potential resources for the breeding of salt-tolerant soybeans [72].

In addition to the introgression of *GmSALT3* into soybeans via marker-assisted selection, new salt-tolerant lines have been created using transgenic approaches. Overexpression lines with the transcription factor *GmSIN1* showed rapid emergence and higher yields compared with the salt-tolerant variety Wei6823 under saline conditions [64]. Soybean seedlings with overexpression of nuclear factor Y C subunit *GmNF-YC14* had higher biomass than the wild type under salt stress [70]. Transgenic soybean lines overexpressing a class B heat shock factor *HSFB2b* had higher survival rates than wild-type Jack after 7 d of 300 mM NaCl treatment, and the variations in the promoter of *HSFB2b* may be useful for breeding tolerant soybeans [68]. By introducing nuclear factor Y subunit *GmNFYA* into soybean Jack, the plant height and survival rate were greatly improved under 300 mM NaCl stress [69].

## 5. Perspective

The salt tolerance of soybeans is the result of contributions from genetic loci involved in different developmental stages. Full-seed germination is the initial step for plants to achieve greater yields in saline fields, especially where the timing of sowing usually depends on rainfall events. Studies have shown that the genetic control of salt tolerance varies at different growth stages in soybeans [13,35,56]. Despite these studies, there is still a gap in knowledge on the genetic responses and underlying mechanisms involved in salt tolerance between the germination and emergence stages. This may be partly due to difficulties in conducting these types of studies, which are time-consuming and labor-intensive. They require screening a large number of accessions for germination and seedling vigor to determine maximum levels of salinity stress on plants and are inherently more difficult under more unpredictable field conditions compared with laboratory conditions [82]. Therefore, it is necessary to develop ways to reduce these complexities and difficulties by developing, for example, feasible selection indicators that can bridge the differences between lab and field environments and high-throughput screening technologies to more accurately and precisely phenotype large numbers of samples, especially when measuring plants grown in control conditions could be avoided. WinRoots is a system recently developed for soybean phenomics study, which made RGB (red–green–blue) images of the roots and shoots canopy phenotype easily collected [83]. Identifying traits related to salt stress responses at particular developmental stages using thermal sensors and RGB imaging will lead to the identification of major QTLs on trait variation that can be applied to breeding. The salt tolerance genes that have been genetically characterized have created opportunities to develop salt-tolerant soybean varieties via marker-assisted selection (MAS) using tightly linked or functional molecular markers.

Salt stress causes a reduction in growth because energetic resources must be allocated away from photosynthetic processes to accommodate the need for osmotic adjustment [4]. The introduction of salt tolerance genes into crops, for example, *Nax2* in wheat and *GmSATL3* in soybean, only mitigates the losses in yield due to salt stress rather than restoring the full yield achievable in non-saline fields due to the energy used for osmotic adjustment [35,51,84]. Therefore, knowing how to more effectively use salt tolerance genes or their regulators in soybean breeding via genetic engineering, i.e., transgenic manipulation or genomic editing, to improve yield under both saline and non-saline conditions depends on our advances in understanding the underlying mechanisms and warrants further exploration. ‘Omics’ approaches may be useful to determine the roles of known genes and identify the pathways and essential genes involved in salt tolerance (Figure 3).

Although we have evaluated some genetic loci, as salt tolerance is a very complex physiological process, the following challenges remain: (1) approaches that can be easily used to screen the phenotypes of soybean accessions; (2) major gene(s) contributing to specific developmental stages without a yield penalty under both saline and non-saline conditions; and (3) effective ways for trait stacking to obtain salt-tolerant soybeans throughout the whole growing period.

## Figures and Tables

**Figure 1 plants-13-00097-f001:**
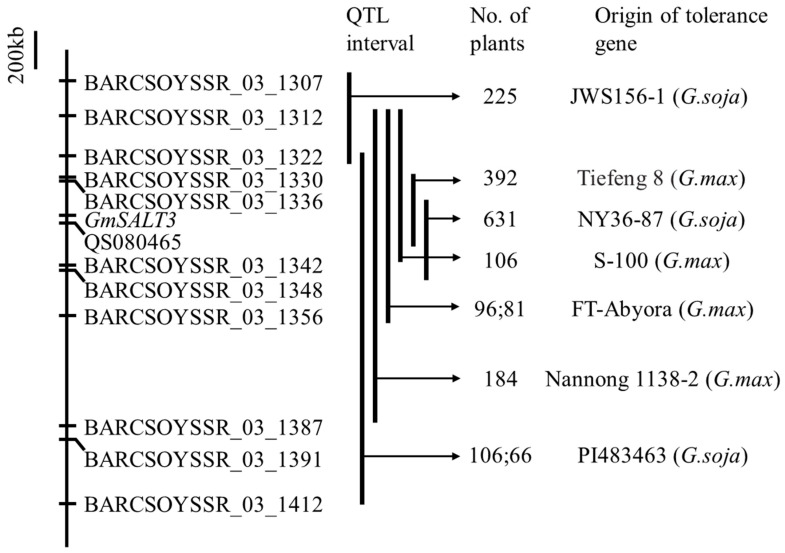
Schematic of salt tolerance QTL/gene mapping interval in Chr. 03 at the seedling stage. Bold vertical lines signify the QTL interval in each study, and the origin of the salt tolerance gene and the number of individuals in the segregating populations are also marked. All markers are *Glyma.Wm82.a2* version [34,46,47,48,49,58,60].

**Figure 2 plants-13-00097-f002:**
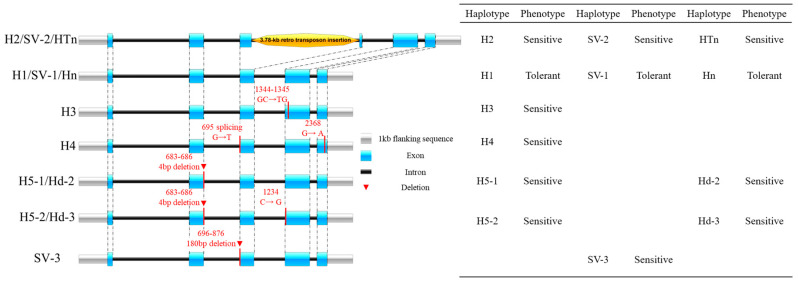
Main structure and coding variations of *GmSALT3* observed in soybean accessions. Blue boxes indicate exons, the black bars represent introns, gray boxes represent the 1 kb flanking sequence of *GmSALT3*, dotted lines indicate the exon position, orange block represents the 3.78 kb retro transposon insertion, and key variants are listed with red letters (only those haplotypes observed in more than 10 accessions were counted) [72,73,74].

**Figure 3 plants-13-00097-f003:**
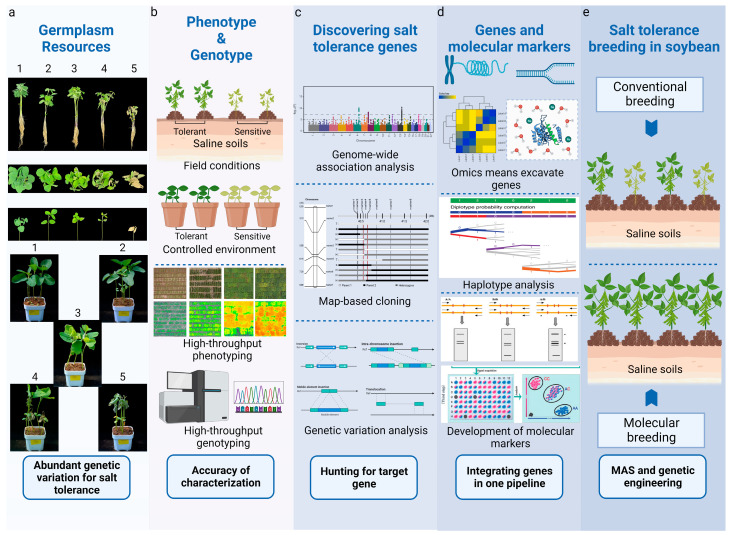
Breeding salt-tolerant soybeans. (**a**) Abundant variations in salt tolerance exist in soybean accessions. Numbers from 1 (the most tolerant) to 5 (the most sensitive) indicate salt tolerance level. (**b**) Characterizing soybean accessions with high-throughput phenotyping methods under conditions that mimic realistic saline field conditions and cost-effective genotyping approaches. (**c**) Identifying target genes and functional variants for salt-tolerance-related traits. (**d**) Integration of single-nucleotide variations and ‘omics’ at gene level. (**e**) With knowledge of key gene variations and related pathways, MAS and other biological approaches will become the most effective way for breeding.

**Table 1 plants-13-00097-t001:** Evaluation and identification of salt-tolerant accessions of soybeans at different developmental stages.

Stage	Stress Condition	Indicators	Total Accessions	No. of Tolerant Accessions	Proportion of Tolerant Accessions	Reference
Germination	1.6% NaCl	Salt damage index	10,128	924	9.1%	Shao et al. (1993) [19]
Germination	2.0% NaCl	Relative salt damage rate	760	141	18.5%	Li et al. (1996) [39]
Germination	150 and 200 mM NaCl	GR	10	3	30.0%	Shelke et al. (2017) [20]
Germination	150 mM NaCl	IR, GR, and GI	191	/	0.0%	Kan et al. (2015) [40]
Germination	1.2% NaCl	Relative salt damage rate	793	117	14.8%	Jiang et al. (2012) [41]
Emergence	Saline soil: 3.1–13.7 dS m^−1^	Relative seedling emergence rate	6	2	33.3%	Abel and MacKenzie (1964) [13]
Emergence	Saline soil: 3–6 dS m^−1^	GR	7	3	42.9%	Wang et al. (1999) [24]
Emergence	150 mM NaCl	SI	27	10	37.0%	Liu et al. (2020) [27]
Seedling	5.0–10.2 dS m^−1^	Cl^–^ content in leaves	6	4	66.7%	Abel and MacKenzie (1964) [13]
Seedling	Saline water (EC = 15–17 dS m^−1^)	Green loss grade (1–5)	10,128	457	4.5%	Shao et al. (1993) [19]
Seedling	25 mM to 150 mM NaCl	SDW	170	18	10.6%	Mannan et al. (2010) [36]
Seedling	120 mM NaCl	Leaf damage	7	3	42.9%	Valencia et al. (2008) [29]
Seedling	100 mM NaCl	Leaf damage	14	5	35.7%	Lee et al. (2008) [30]
Seedling	200 mM NaCl	Leaf damage	8	4	50.0%	Jiang et al. (2013) [32]
Seedling	21 ± 3 dS m^−1^	Salt damage index	793	41	5.17%	Jiang et al. (2012) [41]
Seedling	120 mM NaCl	Leaf damage	98	36	36.7%	Ledesma et al. (2016) [31]
Emergence	200 mM NaCl	SI	27	12	44.4%	Liu et al. (2020) [27]
Whole period	Saltwater irrigation	Degree of salt damage	2000	7	0.4%	Shao et al. (1986) [18]
Whole period	Saline soil	Relative salt tolerance index	793	35	4.41%	Jiang et al. (2012) [41]

## Data Availability

Not applicable.

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
