# Peer review of "Salt Tolerance in Soybeans: Focus on Screening Methods and Genetics"

_plants, 2023, doi:10.3390/plants13010097_

Round 1
Reviewer 1 Report
Comments and Suggestions for Authors
This review evaluates how it has been phenotyped the salt resistance and the identification of genes related to this resistance in soybean.
The abstract could be improved adding a conclusion.
The first paragraphs of the introduction should be reviewed as the references chosen do not correspond with what is said in the text.
I recommend to change germplasm to accession.
Phenotyping is a large word, here you only study the impact of the developmental stage of soybean (which determines the structure of your text) on the resistance to salt, you even do not describe how the plants were considered as resistant vs tolerant or sensitive, or the methods to evaluate it, so you should make it more clear in the title, abstract, introduction and conclusion. You have discussed (but not enough, and not in the discussion) a little the differences between evaluation in field and on pots/hydroponics etc, but this correspond to the support of culture The genetic part is ok.
The last parts are globally ok, perspectives are fine.
I recommend to reorientate the objective of the review to be more adapted to the real development later, find new references to the first paragraphes and correct the minor details that you can find in the document attached

Some sentences and expressions should be corrected, but globally is perfectly understandable
Author Response
Response to Reviewer 1 Comments
- Summary
Thanks for your time handling our manuscript. We have carefully revised the content according the comments. All the changes in manuscript are highlighted.
- Point-by-point response to Comments and Suggestions for Authors
Comments 1: This review evaluates how it has been phenotyped the salt resistance and the identification of genes related to this resistance in soybean. The abstract could be improved adding a conclusion.
Response 1: Thank you for the suggestion. We have added a conclusion in the abstract as “Progress has been made in the past decades, new technologies will help mining of novel salt tolerance genes and translating the mechanism of salt tolerance into new varieties through effective ways” which can be found in line 28–30.
Comments 2: The first paragraphs of the introduction should be reviewed as the references chosen do not correspond with what is said in the text.
Response 2: Thanks for the comments. We checked and added references, revised the first paragraph of the introduction. Detail information is highlighted in the manuscript attached (line 33–47).
Comments 3: I recommend to change germplasm to accession.
Response 3: We changed the word germplasm into accession as the reviewer suggested.
Comments 4: Phenotyping is a large word, here you only study the impact of the developmental stage of soybean (which determines the structure of your text) on the resistance to salt, you even do not describe how the plants were considered as resistant vs tolerant or sensitive, or the methods to evaluate it, so you should make it more clear in the title, abstract, introduction and conclusion. You have discussed (but not enough, and not in the discussion) a little the differences between evaluation in field and on pots/hydroponics etc, but this correspond to the support of culture. The genetic part is ok.
Response 4: Agree. We use ‘screening’ in the revised manuscript instead of phenotyping as the reviewer suggested. We added description of indicators for salt tolerance soybean selection in the related paragraphs of germination (line 66–72), emergence (line 104–107), seedling (line 152–153) and later growing (line 163–165) stages, and the stress conditions and related indicators in different researches were also listed in table 1. To date, the correlation between greenhouse evaluation and performance in the field was observed in screening of salt tolerance at soybean seedling stage, in which the salt tolerance gene GmSALT3 has been isolated through a map-based cloning strategy. This isolation of growth stage specific tolerance gene will then help us to develop effective screening methods that can replicate the variation in germination or emergence phenotypes in the field conditions.
Comments 5: The last parts are globally ok, perspectives are fine.
I recommend to reorientate the objective of the review to be more adapted to the real development later, find new references to the first paragraphes and correct the minor details that you can find in the document attached
Response 5: Thanks for your detailed annotation. We rewrite corresponding parts of the first paragraph and corrected the minor details the reviewer marked in the document, and the revisions are highlighted in attached file (line 79–81, line 92–94, line 317–319).
Comments 6: Some sentences and expressions should be corrected, but globally is perfectly understandable.
Response 6: Thank you so much. All the sentences or expression corrected in the word document are highlighted.

Reviewer 2 Report
Comments and Suggestions for Authors
I read with attention the review entitled: Salt Tolerance in Soybean: focus on Germplasm Phenotyping and Genetics.
In the manuscript, the authors address the complex topic of salinity discussing soybean salt tolerance phenotyping, progress in forward genetics, and potential mechanisms involved in salt tolerance. They then continue with the relevant discussion relating to the importance of translating laboratory findings into field experiments. In the end, they deal with the possibility of developing salt-tolerant soybean varieties able to obtain high and stable yields.
The scope of the review is appropriate for Plants and the study is sufficiently substantial and original to justify its publication.
The paper is generally well written, the presentation is clear and well organized; the summary is concise and the title is suitable and sufficiently descriptive.
All the tables and figures are necessary; the bibliography is representative.
However, the presentation needs to be improved, and an issue requires to be addressed before publication is recommended. A specific point is related to the lack of a description of the effects of salinity on soil chemistry and the impact of salinity on plant metabolism in general. This paragraph helps the reader to better understand the topic which is not easy to understand and is not yet completely known.
Author Response
Response to Reviewer 2 Comments
- Summary
Thanks for your time handling our manuscript. We have carefully revised the content according the comments. All the changes in manuscript are highlighted.
- Point-by-point response to Comments and Suggestions for Authors
Comments 1: The presentation needs to be improved, and an issue requires to be addressed before publication is recommended.
Response 1: Thank you so much for your comments. We have revised the presentation of the manuscript.
Comments 2: A specific point is related to the lack of a description of the effects of salinity on soil chemistry and the impact of salinity on plant metabolism in general. This paragraph helps the reader to better understand the topic which is not easy to understand and is not yet completely known.
Response 2: We added the brief introduction of effects of salinity on soil and possible ways through which it may affect plant growth in the introduction section. All changes in the manuscript are highlighted (line 33–35).

Round 2
Reviewer 1 Report
Comments and Suggestions for Authors
The changes requested have been made, great efforts have been operated on the abstract and in the introduction.
One thing that should be reviewed in all the document is the Accession vs Accessions as there are plenty that have stayed in the singular form as they were replace germplasm, but should be changed to the plural.
I've highlighted several in the attached document.
All the rest if fine

Author Response
Response to Reviewer 1 Comments
- Summary
Thanks for your time handling our manuscript. We changed “accession” from the singular form to the plural form. All the changes in manuscript are highlighted.
We go through the whole manuscript and make some corrections which are all highlighted:
- We delete ‘accessions’ in line 2 and line 353.
- ‘mining of’ is changed to ‘mining’ in line 28.
- ‘of plants’ is delated in line 69.
- ‘a’ is deleted in line 201.
- Point-by-point response to Comments and Suggestions for Authors
Comments 1: One thing that should be reviewed in all the document is the Accession vs Accessions as there are plenty that have stayed in the singular form as they were replace germplasm, but should be changed to the plural.
I've highlighted several in the attached document.
Response 1: Thank you for the careful suggestion. We delete the accession from the title and change 30 “accession” to “accessions”, all of which are highlighted in the manuscript.
